# Recent Progress in III–V Photodetectors Grown on Silicon

Cong Zeng [†], Donghui Fu [†], Yunjiang Jin and Yu Han *

State Key Laboratory of Optoelectronic Materials and Technologies, School of Electronics and Information Technology, Sun Yat-Sen University, Guangzhou 510275, China; zengc3@mail2.sysu.edu.cn (C.Z.)
* Correspondence: hany87@mail.sysu.edu.cn
† These authors contributed equally to this work.

**Abstract:** An efficient photodetector (PD) is a key component in silicon-based photonic integrated circuits (PICs). III–V PDs with low dark current density, large bandwidth, and wide operation wavelength range have become increasingly important for Si photonics in various applications. Monolithic integration of III–V PDs on Si by direct heteroepitaxy exhibits the lowest cost, the largest integration density, and the highest throughput. As the research of integrating III–V lasers on Si flourishes in the last decade, various types of III–V PDs on Si with different device structures and absorption materials have also been developed. While the integration of III–V lasers on Si using various technologies has been systematically reviewed, there are few reviews of integrating III–V PDs on Si. In this article, we review the most recent advances in III–V PDs directly grown on Si using two different epitaxial techniques: blanket heteroepitaxy and selective heteroepitaxy.

**Keywords:** Si photonics; III–V photodetector; blanket heteroepitaxy; selective heteroepitaxy

## 1. Introduction

Si photonics is an attractive platform for PICs in numerous emerging applications due to its unique advantages, such as low cost, large integration density, and mature manufacturing processes [1,2]. Key devices such as lasers, modulators, and PDs have been demonstrated with impressive performances in commercial Si photonic platforms [3–5]. Among the key components in Si photonics, PDs play an important role in converting optical signals into electrical ones, thus connecting PICs with ICs. Typically working under reverse bias, PDs absorb incident photons and transform them into electron-hole pairs, which are separated by the large electric field and subsequently collected by the metal pads [6]. In addition to the general metrics, namely dark current, responsivity, and bandwidth, integrating PDs in Si photonic platforms not only necessitates waveguide coupling instead of free-space coupling but also requires the PDs to be integrated in a low-cost and scalable manner.

Ge PDs have been widely deployed in Si photonics, benefiting from their decent responsivity, large bandwidth, and compatibility with the complementary metal oxide semiconductor (CMOS) manufacturing lines [7,8]. However, as the applications of Si photonics extend from datacom/telecom communication towards high-performance computing [9], display [10], and sensing [11], conventional Ge PDs can hardly fulfill the different requirements in these diverse application scenarios. Firstly, Ge is an indirect bandgap semiconductor with a relatively small absorption coefficient; secondly, the dark current of Ge PDs grown on Si is large due to the existing crystal defects and the inherent large thermionic emission coefficient [7]; thirdly, the absorption spectrum of Ge drops considerably beyond 1550 nm which precludes its application in longer wavelengths. In contrast, III–V PDs based on direct bandgap semiconductors feature large absorption coefficients, and the flexible band engineering of these binary, ternary, and quaternary compounds engenders a variety of high-performance PDs with operating wavelengths spanning from the near-infrared all the way to the mid-infrared band [12,13]. In addition, as the on-chip lasers of Si photonics

are fabricated from III–V compound semiconductors, III–V PDs can also be integrated into the Si photonics platform in a similar manner [14,15].

Similar to the integration of III–V lasers on Si [3,16], both heterogeneous integration and monolithic integration have been employed to integrate III–V PDs on Si [5]. Using wafer/die bonding and transfer printing, heterogeneous integration has demonstrated III–V PDs on Si with performance comparable and even superior to those on native III–V substrates [17–19]. This is because heterogeneous integration seamlessly combines the light absorption/emission ability of III–V compounds and the excellent waveguiding capability of Si. However, heterogeneous integration features high cost and small throughput, which compromises its potential application in consumer-oriented photonics, where cost is the primary concern. As an alternative, monolithic integration using direct heteroepitaxy provides a low-cost and high-throughput solution [20]. However, it is by no means a trivial task to directly grow III–V compounds on Si as mismatches in lattice constant, polarity, and coefficient of thermal expansion give rise to crystal defects such as threading dislocations (TDs) and antiphase boundaries (APBs) [21,22]. Both TDs and APBs serve as current leakage paths and cause a sharp increase in the dark current of the epitaxial PDs. APBs with electrically charged III–III and V–V bonds could also lead to charge accumulation and impair the responsivity and the high-frequency performance of III–V PDs grown on Si [21]. Thanks to the intense research of III–V heteroepitaxy in the past decade, the issues of APBs have been completely resolved, and the density of TDs has also been drastically reduced [22]. Additionally, unlike lasers operating in the forward-biased mode with intense light-matter interaction and extreme sensitivity to the influence of TDs [23], most PDs operate in the reverse-biased mode and are, therefore, much less sensitive to the influence of TDs.

In general, currently, there are two technologies to grow III–V compound semiconductors on Si substrates: blanket heteroepitaxy and selective heteroepitaxy. Blanket heteroepitaxy directly deposits III–V thin films on large-scale Si wafers, while selective heteroepitaxy produces III–V materials in localized regions of pre-patterned Si substrates [24,25]. Table 1 compares the pros and cons of III–V PDs grown on Si using these two technologies. Since blanket heteroepitaxy yields III–V/Si compliant substrates, the subsequent growth and fabrication of PDs are identical to that on native III–V substrates, and thus blanket heteroepitaxy is an extremely versatile technology with the demonstration of a variety of PDs [5,12,26]. However, the thick buffer layer necessary for defect reduction impedes the light coupling between the epitaxial III–V PDs and the Si waveguides. As a result, it is challenging to seamlessly integrate III–V PDs on the Si photonics platform using blanket heteroepitaxy. In contrast, the inherent defect necking effect of selective heteroepitaxy guarantees an intimate placement of the epitaxial III–V materials with the Si substrates and accordingly leads to efficient light coupling between the epitaxial III–V PDs and the Si waveguides. Several high-performance III–V PDs have recently been demonstrated in the Si photonics platform using selective heteroepitaxy [27,28].

**Table 1.** Comparison of blanket heteroepitaxy and selective heteroepitaxy of III–V PDs.

|  | Blanket Heteroepitaxy | Selective Heteroepitaxy |
|---|---|---|
| TD density | GaAs/Si~$10^6$ cm$^{-2}$<br>InP/Si~$10^8$ cm$^{-2}$<br>GaSb/Si~$10^7$ cm$^{-2}$ | Potentially TD free |
| Dark current density | ~$10^{-7}$–$10^0$ A/cm$^2$ | ~$10^{-8}$–$10^{-2}$ A/cm$^2$ |
| Speed | ~1.5–28 GHz | ~1.1–70 GHz |
| Wavelength | GaAs-based: 850~1310 nm & around 6 µm<br>InP-based: 1310~1550 nm<br>GaSb-based: 2~6.4 µm | 1200~1700 nm |
| PD Types | Almost all types of III–V PDs | Mainly PIN |
| Si-Waveguide PDs | Inefficient butt coupling | Efficient butt and evanescent coupling |

The integration technologies of III–V lasers and PDs are quite similar. While there are several excellent review articles on integrating III–V lasers on Si using heterogeneous integration as well as monolithic integration [3,16,25], an extensive review of monolithically integrating III–V PDs on the Si photonics platform using direct heteroepitaxy is still lacking. In this article, we review the most recent progress of III–V PDs directly grown on Si substrates and highlight their potential for seamless integration with Si photonics.

## 2. Blanket Heteroepitaxy of III–V PDs on Si

The research of blanket heteroepitaxy dates back to the early 1980s and was initially targeted at developing efficient III–V electronic devices on Si [29]. As the investigation of Si photonics boomed in the 2010s, the search for on-chip light sources reanimated the studies of blanket heteroepitaxy [30]. Several major breakthroughs have been achieved in the past decade; for instance, GaAs-based quantum dot lasers grown on Si could operate reliably in high-temperature ambiance, and this technology is now being commercialized [31]. As both the epitaxy and device designs of III–V photonics can be readily applied in blanket heteroepitaxy, almost all types of III–V PDs can be fabricated on III–V/Si compliant substrates without major structural adjustments. Examples include vertical p-i-n photodiodes (VPIN PDs), avalanche photodiodes (APDs), modified uni-traveling carrier photodiodes (MUTC PDs), and barrier-design photodiodes (nBn PDs).

The investigation of blanket heteroepitaxy has been centered on eradicating APBs and reducing the TD density of the epitaxial III–V thin films. Targeting applications in the near-infrared, telecom, and mid-infrared bands, GaAs, InP, and GaSb thin films have all been successfully grown on Si wafers without APBs. Due to the difference in lattice mismatch and growth dynamics, the TD density of GaAs/Si thin film has been reduced to $10^6$ cm$^{-2}$, while that of InP/Si and GaSb/Si are in the order of $10^8$ cm$^{-2}$ and $10^7$ cm$^{-2}$, respectively [32–34]. Figure 1 lists the basic parameters of III–V PDs grown on Si based on GaAs, InP, and GaSb material platforms. In this section, we first briefly introduce the basic techniques used to exterminate APBs and lower the density of TDs, then review the recent progress of GaAs-based, InP-based, and GaSb-based PDs grown on Si, and finally discuss the possibilities of coupling light from Si waveguides in the Si photonics platform. The performance parameters of different types of III–V PDs integrated on Si using blanket heteroepitaxy can be found in Table 2.

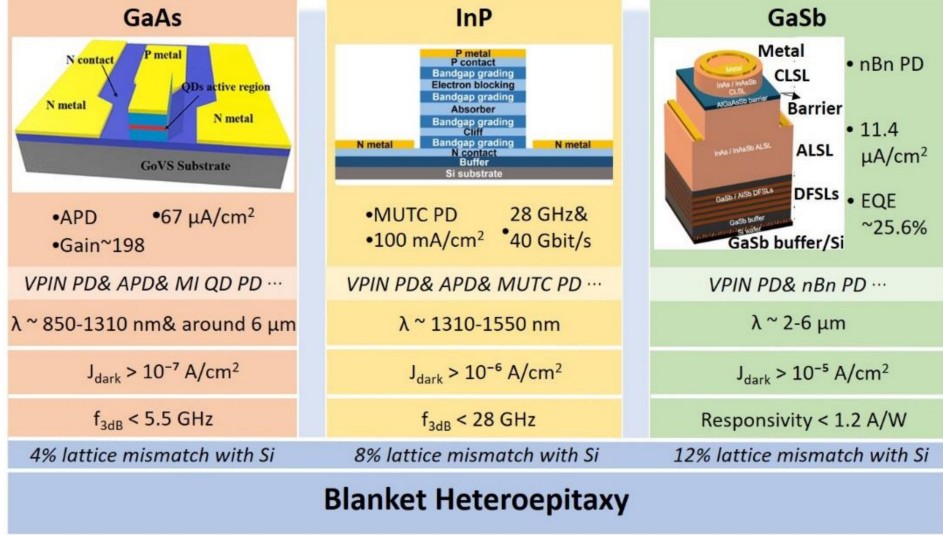

**Figure 1.** Basic parameters of III–V PDs grown on Si by blanket heteroepitaxy and the representative PDs for different material platforms: APD for GaAs-based. Reprinted with permission from Ref. [35]. Copyright 2020, American Chemical Society; MUTC PD for InP-based [36]. And nBn PD for GaSbbased. Reprinted with permission from Ref. [34]. Copyright 2020, Society of Photo-Optical Instrumentation Engineers (SPIE).

### 2.1. Defect-Management Techniques

Given that the growth of InP and GaSb thin films on Si often start from GaAs/Si compliant substrates and the defect-management mechanisms are quite similar for the three compounds, we chose GaAs as an example to illustrate the basic defect reducing techniques as schematically illustrated in Figure 2. Earlier studies often adopted Si substrate with an offcut angle of 4–6° to preclude the formation of APBs because these offcut wafers feature double-atomic steps, while on-axis (001) Si wafers exhibit single-atomic steps [37,38]. However, the Si photonics industry uses (001) Si wafers with offcut angles smaller than 0.5° and, as a consequence, other strategies have been developed to remove APBs in III–V thin films grown on on-axis (001) Si wafers as shown in Figure 2a. First, GaP could serve as an intermediate buffer between GaAs and Si, and APBs could be completely annihilated within a thin GaP layer [39]. This technology was developed by NAsP$_{\mathrm{III-V}}$, and currently, 300 mm GaP/Si templates are commercially available. Second, with high-temperature annealing and careful tuning of the low-temperature nucleation layer, APBs can also be extirpated in GaAs thin films directly grown on Si without any intermediate buffers [40]. Third, patterning V-grooves on (001) Si wafers could also eliminate APBs as the anisotropic-etched {111} facets exhibit double-atomic steps and prevent the formation of APBs in the nucleation stage [41].

In addition to APBs, a TD density of up to $10^9$ cm$^{-2}$ usually exists inside the epitaxial III–V thin films without special treatments. Figure 2b lists several representative approaches to further reduce the TD density. The most straightforward approach is to grow a thicker III–V buffer, such as GaAs or Ge, which promotes the self-annihilation of TDs [21]. However, such a thick buffer layer will inevitably bring other problems, such as cracks induced by the difference in thermal expansion coefficient and increased cost of the precursors and production time. Therefore, other TD reducing approaches, including thermal cycle annealing (TCA), strained layer superlattices (SLS), and misfit trapping layers (MTL), have been developed accordingly [31,42,43]. The TCA process activates the motion of TDs and increases the probability of annihilation. The induced strain of SLS could bend the propagation of TDs, thus preventing them from reaching the crystal surface. Sandwiching the active region with MTL could effectively prevent the formation of misfit dislocations in the active region and thus significantly improve the device's reliability. Employing the combination of these TD reducing methods, the TD density of epitaxial GaAs thin films on Si has been reduced to an impressive level of just $10^6$ cm$^{-2}$.

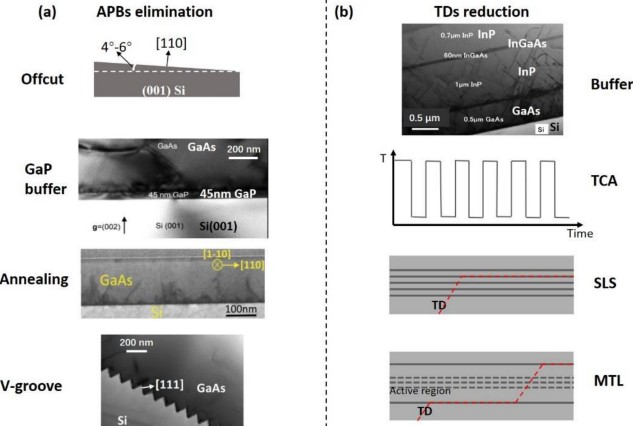

**Figure 2.** Defect-management techniques. (**a**) APB elimination: offcut angle of 4–6°; GaP intermediate buffer. Reprinted with permission from Ref. [44]. Copyright 2017, AIP Publishing; high-temperature annealing and tuning of the low-temperature nucleation layer. Reprinted with permission from Ref. [40]. Copyright 2016, AIP Publishing; V-groove with anisotropic-etched 111 facets. Reprinted with permission from Ref. [45]. Copyright 2015, AIP Publishing. (**b**) TDs reduction: thick buffer. Reprinted with permission from Ref. [46]. Copyright 2017, Elsevier; thermal cycle annealing; strained layer superlattices; misfit trapping layers.

### 2.2. GaAs-Based III–V PDs

GaAs-based PDs traditionally operate in the near-infrared region. With the introduction of InAs quantum dots (QDs) as the active medium, GaAs-based PDs can also operate in the 1.3 μm band [32]. The detecting wavelength can be further extended to the mid-infrared region by utilizing the intra-subband transitions inside the InAs QDs [47]. Figure 3 depicts the schematics of three types of PDs with different device structures, including VPIN PDs, APDs, and unipolar mid-infrared PDs. Compared to InP and GaSb, GaAs has a smaller lattice mismatch with Si, and the TD density of the GaAs/Si-compliant substrates is generally an order of magnitude smaller. As a result, the dark current of GaAs-based PDs grown on Si is often fairly low and sometimes comparable to that grown on native GaAs substrate. Mehdi et al. demonstrated GaAs VPIN PDs grown on Ge/Si substrates, which use GaAs as the active region and exhibit a low dark current density of 0.45 μA/cm$^2$ at a bias voltage of −2 V and a responsivity of 0.17 A/W at 850 nm [48]. This performance is comparable to identical epitaxial structures grown on GaAs substrate. As the investigation of InAs QD lasers flourished in the last decade [49–51], GaAs-based VPIN PDs grown on Si with InAs QDs as the active gain medium have also been studied [32]. Compared with quantum wells, the stronger carrier confinement of QDs significantly reduces the current leakage from TDs and device sidewalls [52]. Figure 3a depicts the epitaxial structure of VPIN PDs with InAs QDs absorption layers grown on GaP/Si substrates. These devices show a low dark current density of 0.13 mA/cm$^2$ at a bias voltage of −3V, and a 10 Gbit/s eye opening is confirmed.

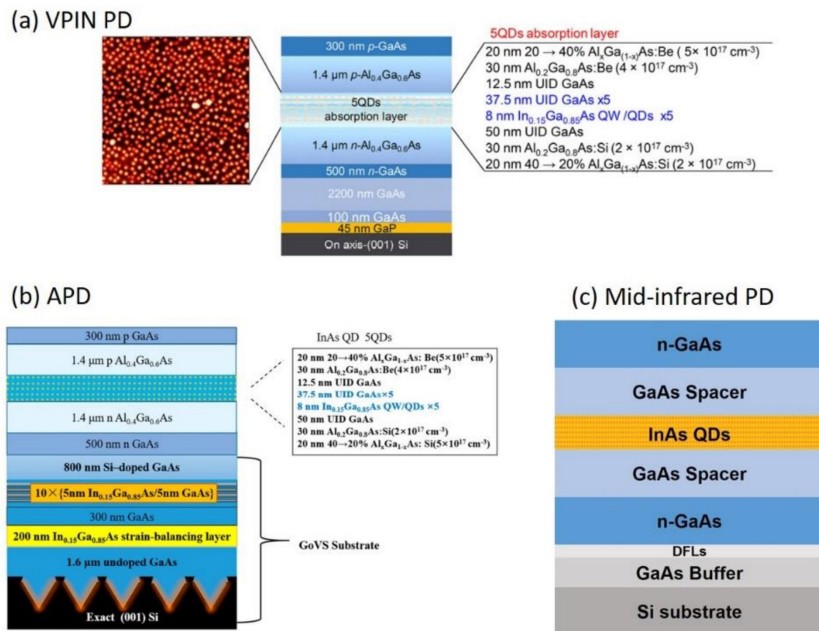

**Figure 3.** Main types of GaAs-based III–V PDs: (**a**) VPIN PD. Reprinted with permission from Ref. [32]. Copyright 2018, AIP Publishing; (**b**) APD. Reprinted with permission from Ref. [35]. Copyright 2020, American Chemical Society; (**c**) unipolar mid-infrared PD [53].

In addition to conventional VPIN PDs, other types of PDs with complex band structure engineering can also be grown on the GaAs/Si-compliant substrate. Chen et al. demonstrated the first O-band InAs QD APDs grown on Si with the device structure shown in Figure 3b [35]. Five-stacked InAs dot-in-a-well (DWELL) structures with a dot density of $6 \times 10^{10}$ cm$^{-2}$ serve as both an absorption region and a multiplication region in this APD. This structure was grown on a GaAs-on-V-grooved-Si (GoVS) substrate to reduce the TD density, and the sidewall is passivated with a 12 nm Al$_2$O$_3$ together with a 1 μm SiO$_2$ layer to suppress the surface leakage current. The device exhibits a dark current density as low as 67 μA/cm$^2$ when biased at −5 V, and the low dark current voltage (I–V) curves are

shown in Figure 4a. Figure 4b plots the evolution of the gain as a function of bias, and a maximum gain of 198 is obtained at 293 K. These APDs with low dark current and high gain could play an important role in Si-based transceiver applications. Such APDs use the same epitaxial layers and fabrication process as QD lasers, but the authors note that the reuse of the QD layer as both absorption region and multiplication region causes a high excess noise considering the mixed carrier injection [54].

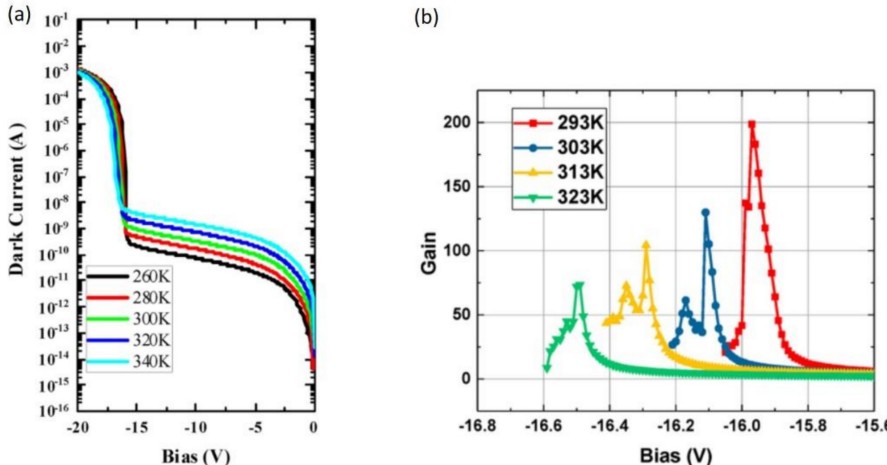

**Figure 4.** Temperature-dependent measurement of the (**a**) current-voltage and (**b**) gain-voltage characteristics of the APDs grown on Si. Reprinted with permission from Ref. [35]. Copyright 2020, American Chemical Society.

Thanks to the three-dimension quantum confinement of QDs and the resultant discrete density of states, intra-subband transitions can be utilized to fabricate PDs operating at the mid-infrared wavelength region. Figure 3c displays the detailed device structure of mid-infrared GaAs-based PDs grown on Si. Different from traditional PIN structures, most of these QD PDs have two n-type contact layers on both sides of the active region. Chen et al. demonstrated the first mid-infrared InAs/InGaAs/GaAs DWELL PDs on Si [53]. The use of dislocation filter layers (DFLs) reduces the TD density to $3 \times 10^6/\text{cm}^2$, and the DWELL active region is subsequently sandwiched between two n-type GaAs contact layers atop the DFLs. The devices exhibit an ultra-low dark current density of 2.03 $\mu\text{A/cm}^2$ under 1 V bias at 77 K and a peak specific detectivity of $5.78 \times 10^8$ cm·Hz$^{1/2}$/W under 2 V bias at 6.4 $\mu$m and 77 K, corresponding to a responsivity of 10.9 mA/W.

*2.3. InP-Based III–V PDs*

InP and its related compounds are the dominant materials for optoelectronic devices operating in the telecom band, especially in the technologically important 1550 nm band. In contrast to GaAs, InP and Si have a larger lattice mismatch of up to 8%, which brings a greater challenge for the blanket heteroepitaxy of InP on Si. TD density in the order of $10^8$ cm$^{-2}$ is common for InP thin films grown on Si, and the dark current is thus generally larger than that grown on GaAs/Si templates [33,55]. Figure 5 lists several representative types of InP-based PDs grown on Si, including VPIN PDs, APDs, and MUTC PDs. In addition to using InGaAs as the absorption material [56], InP-based VPIN PDs could also reduce the leakage current by employing quantum dashes (QDash) and QDs as the absorption material [57]. Figure 5a displays InP-based QDash VPIN PDs directly grown on (001) Si. The VPIN structure with a QDash active region is grown on an optimized InP buffer atop the GaAs/Si-compliant substrates [33]. A low dark current density of 2.1 $\mu\text{A/cm}^2$, responsivities of 0.35 A/W at 1550 nm and 0.94 A/W at 1310 nm, and a 3 dB bandwidth of 10.3 GHz are demonstrated. Unlike GaAs-based QD PDs operating in the O-band, these InP-based QDash PDs can operate in the entire telecom band, spanning from 1240 nm to 1640 nm.

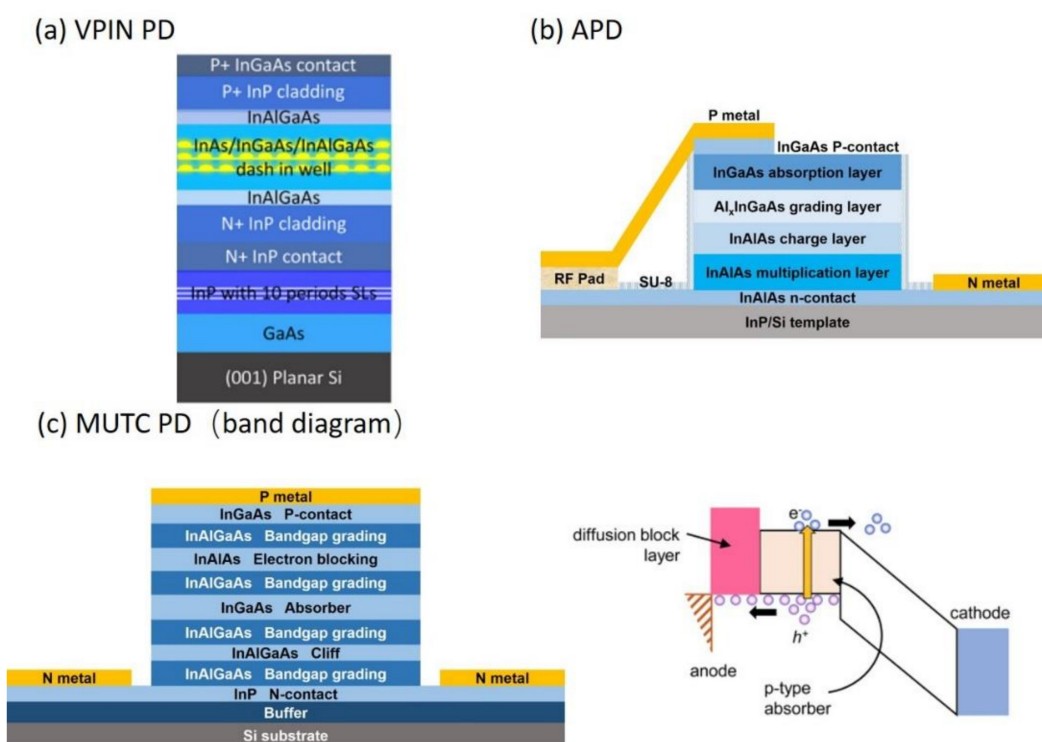

**Figure 5.** Main types of InP-based III–V PDs: (**a**) VPIN PD. Reprinted with permission from Ref. [33]. Copyright 2021, AIP Publishing; (**b**) APD [58]; (**c**) MUTC PD (left) [36] and the schematic of band diagram (right). Reprinted with permission from Ref. [59]. Copyright 2020, AIP Publishing.

Apart from VPIN PDs, high-gain, high-efficiency III–V APDs operating at 1550 nm have also been demonstrated on InP/Si templates. Yuan et al. demonstrated the first III–V APDs directly grown on InP/Si templates [58], and the device structure is detailed in Figure 5b. The InP/Si template consists of Ge/GaAs/InAlAs buffer layers, and the authors chose separate absorption, charge, and multiplication (SACM) structures with an InGaAs absorber and an InAlAs multiplication region. The fabricated APDs exhibit a low dark current density <1 $\mu$A/cm$^2$, gain >20, and external quantum efficiency >40%. Such APDs can be deployed in light detection and ranging (LIDAR) [60], quantum photonic circuits [26], and future high-bandwidth-density interconnect applications [61].

In addition to VPIN PDs and APDs, MUTC PDs with high speed and high-power handling capabilities have also been directly grown on InP/Si templates. As electron mobility is an order of magnitude higher than whole mobility, MUTC PDs can achieve a much higher speed than their PIN counterparts by restricting hole transport with a diffusion block layer and solely relying on electron transport. Furthermore, the high electron velocity reduces the field modulation by the space charges in MUTC PDs, which promises a higher output saturation level and the resultant high-power handling capability [59]. Sun et al. demonstrated MUTC PDs directly grown on Si, and Figure 5c shows the epitaxial structure [36]. The absorber layers are composed of a p-type graded-doped InGaAs-undepleted region and an n-type InGaAs depleted area as the diffusion block layer. The PDs have a dark current density of 100 mA/cm$^2$ at −3 V, an internal responsivity of 0.78 A/W, and a 3 dB bandwidth of 28 GHz.The open eye diagrams can be detected at 40 Gbit/s in the optical system. Meanwhile, the RF power under −5 V bias is −0.6 dBm at −0.5 dB compression, which attests to the excellent high-power characteristics of the heteroepitaxial III–V PDs.

*2.4. GaSb-Based III–V PDs*

Together with other compounds in the 0.6 nm lattice family, GaSb has been the dominant material platform for light emission and detection in the mid-infrared region. However, with a lattice mismatch up to 12% between GaSb and Si, blanket heteroepitaxy of

GaSb on Si often results in a TD density of $10^7$ cm$^{-2}$. Interestingly, this TD value is still lower than InP/Si templates, probably due to the unique interfacial misfit (IMF) formed in the heteroepitaxy of GaSb [62]. The dark current of GaSb-based PDs is often much larger than their GaAs and InP counterparts due to the narrower band gap. Unlike GaAs and InP PDs grown on Si with systematic and extensive studies, the research on GaSb-based PDs directly grown on Si is quite limited. Figure 6 shows the device structures of two main kinds of GaSb-based PDs on Si: VPIN PD and nBn PD, with nBn standing for an n-type absorption layer, a barrier layer, and an n-type contact layer. The GaSb-based VPIN PD is mainly based on InAs/GaSb type-II superlattices (T2SL). Burguete et al. demonstrated the first T2SL VPIN PDs directly grown on Si substrates in 2018 [63]. The dark current density of the devices is 10 A/cm$^2$ under $-0.1$ V bias because of the significant leakage through dislocations and surface states. Deng et al. demonstrated T2SL VPIN PDs on Si with better performance, as schematically shown in Figure 6a [64]. By optimizing the III/V-Si interface and material growth quality, the TD density has reduced to around $2.6 \times 10^8$ cm$^{-2}$. Such a device exhibits a dark current density of 2.3 A/cm$^2$ under $-0.1$ V bias at 70 K, peak responsivity of 1.2 A/W, and specific detectivity of $1.3 \times 10^9$ cm·Hz$^{1/2}$/W.

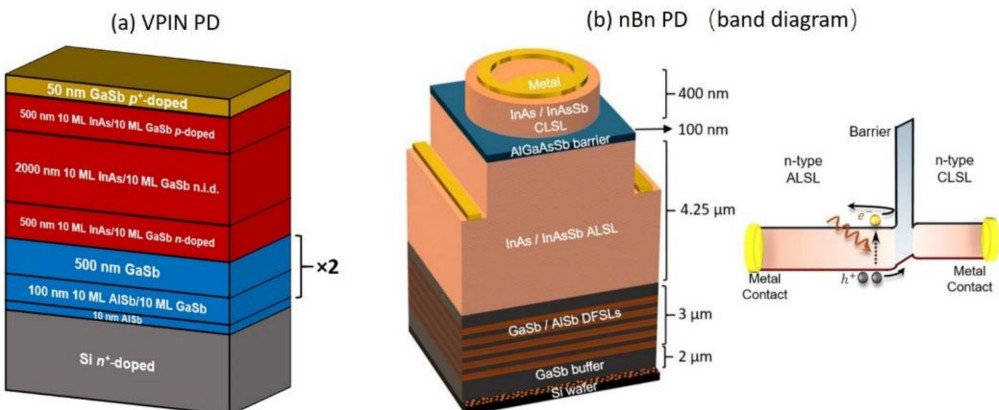

**Figure 6.** Main types of GaSb-based III–V PDs: (**a**) VPIN PD. Reprinted with permission from Ref. [64]. Copyright 2019, Elsevier; (**b**) nBn PD (**left**). Reprinted with permission from Ref. [34]. Copyright 2020, Society of Photo-Optical Instrumentation Engineers (SPIE). And the schematic of band diagram (**right**). Reprinted with permission from Ref. [65]. https://doi.org/10.1021/acsphotonics.8b01550 (accessed on 25 February 2023). Copyright 2019, American Chemical Society.

In contrast to VPIN PDs, nBn PDs exhibit lower dark current density and higher quantum efficiency and have also been demonstrated on Si substrates. As shown by the band structure in Figure 7a, nBn PDs contain a wide-bandgap barrier layer sandwiched between the narrow bandgap absorption and contact layers [66]. A negligible valence band energy offset between the barrier and absorption layer allows the free passage of minority carriers, while a large conduction band energy offset blocks the flow of majority carriers. Through the near elimination of the electric field in the narrow bandgap material and the junction-related Shockley-Read-Hall (SRH) dark current, nBn PDs feature higher operating temperatures and less sensitivity to crystalline defects than conventional VPIN PDs [67]. Carrington et al. demonstrated high-performance InAs/InAsSb nBn PDs directly grown on Si, and Figure 6b presents the device structure [34]. A low dark current density of 11.4 µA/cm$^2$ under $-0.1$ V bias at 110 K is demonstrated, which is much lower than that of GaSb-based VPIN PDs grown on Si. Figure 7b shows the spectral responsivity and external quantum efficiency as a function of temperature. Such PDs exhibit a maximum responsivity of 0.88 A/W at 200 K, corresponding to an external quantum efficiency of 25.6%.

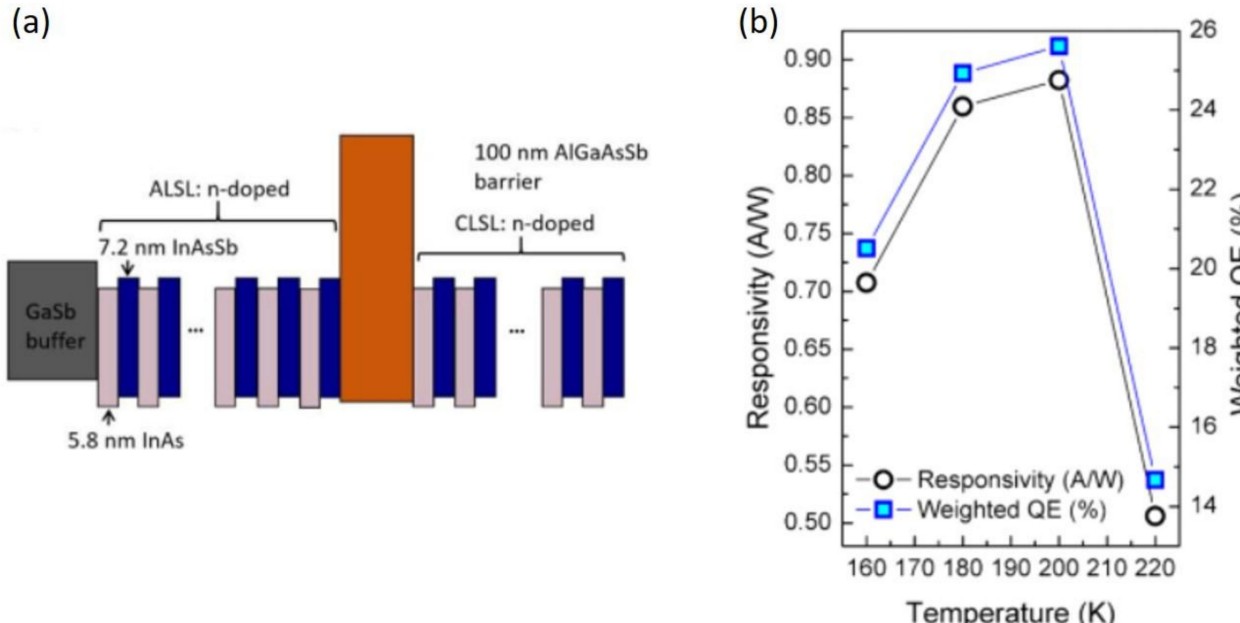

**Figure 7.** (**a**) Schematic diagram and (**b**) spectral responsivity and external quantum efficiency measurements of the nBn PDs grown on Si. Reprinted with permission from Ref. [34]. Copyright 2020, Society of Photo-Optical Instrumentation Engineers (SPIE).

### 2.5. Si Waveguide-Coupled III–V PDs

In the discussions above, all the III–V PDs grown on Si are demonstrated as stand-alone devices, and none are light-coupled using Si waveguides. This is because the buffer layers, up to a few microns thick, impede the light coupling between Si waveguides underneath and the above epitaxial III–V photonic devices. However, to be seamlessly integrated into the current Si photonics platform, coupling with Si waveguides is a prerequisite. In addition, as the light propagation in Si waveguides is often perpendicular to the carrier transport direction in the epitaxial PDs, Si waveguide-coupled III–V PDs could simultaneously obtain high responsivity and large bandwidth. Strategies for light coupling between epitaxial III–V photonic devices and Si waveguides have been proposed and theoretically verified [68,69]; however, experimental demonstrations are quite limited. Figure 8a presents the schematic of Si-waveguide butt-coupled III–V PDs directly grown on SOI substrates. In the demonstration by Feng et al. in 2012, the InGaAs PIN PDs are grown inside an etched well with an area of 5 mm × 15 mm [70]. III–V growth was initiated from the SOI substrate while the Si waveguide was defined at the upper Si device layer. The PDs exhibit a dark current density of 625 mA/cm$^2$ at −1 V, a responsivity of 0.22 A/W, and a 3dB bandwidth of 9 GHz at −4 V. Note that although the III–V PDs were selectively grown at predefined areas on the SOI substrate, the area size is too large to induce any defect necking effect and the growth techniques are also identical to those of blanket heteroepitaxy. Therefore, we categorize this demonstration as blanket heteroepitaxy and present the discussion here. However, due to the inevitable loading effect during growth, the light coupling efficiency between the Si waveguide and the epitaxial III–V PD is still quite low. In such demonstration, the gap between the Si waveguide and the InGaAs absorption layer is around 2 μm, and the InGaAs mesa exhibits a large slope angle ranging from 45 to 60°. A coupling efficiency of just 30% was estimated by the authors.

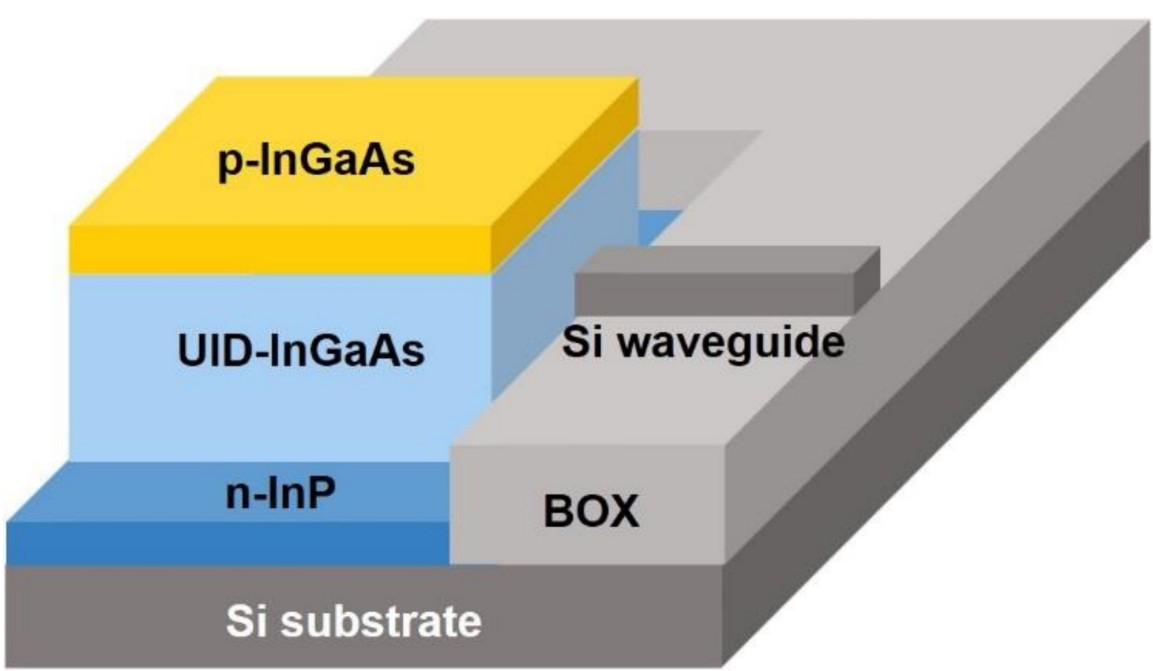

**Figure 8.** Si waveguide-coupled III–V PDs grown on Si by blanket heteroepitaxy [70].

**Table 2.** Summary of blanket heteroepitaxy III–V PDs on Si.

| Year | Material Based | Type | Buffer | Absorption Material | TD Density | Wavelength | Dark Current Density | Responsivity | Speed | Ref. |
|---|---|---|---|---|---|---|---|---|---|---|
| 2012 | GaAs | VPIN | GaAs | InAs/InGaAs DWELL | - | 1280 nm | ~10 µA/cm$^2$ (−1 V) | 5 mA/W | - | [71] |
| 2017 | GaAs | VPIN | GoVS | InAs/InGaAs QD | $7 \times 10^7$ cm$^{-2}$ | 1300 nm | 0.08 mA/cm$^2$ (−1 V) | 0.94 A/W (−1 V) | 1.5 GHz | [72] |
| 2018 | GaAs | VPIN | GaP/GaAs | InAs/InGaAs QD | $8.4 \times 10^6$ cm$^{-2}$ | 1280 nm | 0.13 mA/cm$^2$ (−3 V) | 0.23 A/W | 5.5 GHz (−5 V) 10 Gbit/s | [32] |
| 2020 | GaAs | VPIN | Ge | GaAs | $3.3 \times 10^7$ cm$^{-2}$ | 850 nm | 0.45 µA/cm$^2$ (−2 V) | 0.17 A/W (−2 V) | - | [48] |
| 2021 | GaAs | VPIN | Ge | InGaAs/AlGaAs QW | $4 \times 10^7$ cm$^{-2}$ | 940 nm | 25 µA/cm$^2$ (−1 V) | 36 mA/W (−2 V) | - | [73] |
| 2020 | GaAs | APD | GoVS/InGaAs | InAs DWELL | - | 1310 nm | 67 µA/cm$^2$ (−5 V) | 0.234 A/W (−5 V) | 2.26 GHz 8 Gbit/s | [35] |
| 2016 | GaAs | QDIP | GaAs | InAs/GaAs QD | ~10$^6$ cm$^{-2}$ | 5–8 µm | 0.89 mA/cm$^2$ (1 V, 69 K) | - | - | [47] |
| 2018 | GaAs | QD-QCD | GaAs | InAs/GaAs QD | - | 6200 nm | 21.1 nA/cm$^2$ (−0.1 V, 77 K) | 0.59 mA/W (77 K) | - | [74] |
| 2018 | GaAs | QDIP | GaAs | InAs/InGaAs/ GaAs DWELL | ~$3 \times 10^6$ cm$^{-2}$ | 6400 nm | 2.03 µA/cm$^2$ (1 V, 77 K) | 10.9 mA/W (2 V, 77 K) | - | [53] |
| 2018 | GaAs | QDIP | GaAs | InAs/GaAs QD | ~$3 \times 10^8$ cm$^{-2}$ | 6200 nm | 3.2 A/cm$^2$ (0.2 V, 80 K) | 27 mA/W (0.6 V, 32 K) | - | [75] |
| 2023 | GaAs | QD-QCD | GaAs | InGaAs/GaAs QD | ~10$^7$ cm$^{-2}$ | 6000 nm | ~1 mA/cm$^2$ (−0.1 V, 77 K) | - | - | [76] |
| 1996 | InP | MSM | GaAs/InP | InGaAs | $2 \times 10^7$ cm$^{-2}$ | 1300 nm | 4–40 mA/cm$^2$ (5 V) | 0.26 A/W (5 V) | 1.5 GHz (5 V) | [77] |
| 2012 | InP | VPIN | GaAs/InP | InGaAs | - | 1550 nm | 64 mA/cm$^2$ (−1 V) | 0.57 A/W (−5 V) | 5 GHz (−1 V) | [78] |
| 2012 | InP | VPIN | GaAs/InP | InGaAs | - | 1550 nm | 625 mA/cm$^2$ (−1 V) | 0.17 A/W (−1 V) | 9 GHz (−4 V) | [70] |

**Table 2.** *Cont.*

| Year | Material Based | Type | Buffer | Absorption Material | TD Density | Wavelength | Dark Current Density | Responsivity | Speed | Ref. |
|------|------|------|--------|---------------------|-----------|------------|----------------------|--------------|-------|------|
| 2014 | InP | VPIN | GaAs/InP | InGaAs | - | 1550 nm | 12 mA/cm$^2$ (−1 V) 40 mA/cm$^2$ (−1 V) | 0.02 A/W (−1 V) 0.6 A/W (−1 V) | 14 GHz (−5 V) 15 GHz (−5 V) | [56] |
| 2017 | InP | VPIN | Ge/GaAs/ InAlAs | InGaAs | - | 1550 nm | 1.3 mA/cm$^2$ (−3 V) | 0.76A/W | 8 GHz (3 V) | [55] |
| 2021 | InP | VPIN | GaAs/InP | InAs/InGaAs/ InAlGaAs QDash in well | $3.6 \times 10^8$ cm$^{-2}$ | 1550 nm 1310 nm | 2.1 μA/cm$^2$ (−1 V) | 0.35 A/W 0.94 A/W | 10.3 GHz (−5 V) | [33] |
| 2022 | InP | VPIN | GoVS/InP | InGaAs | $2 \times 10^8$ cm$^{-2}$ | 1550 nm | 0.45 mA/cm$^2$ (−1 V) | 0.72 A/W | 11.2 GHz (−8 V) | [79] |
| 2019 | InP | APD | Ge/GaAs/ InAlAs | InGaAs | - | 1550 nm 1310 nm | <1 μA/cm$^2$ (−5 V) | ~0.54 A/W (−15 V) ~0.48 A/W (−15 V) | - | [58] |
| 2018 | InP | MUTC | Ge/GaAs/ InAlAs/ InP | InGaAs | - | 1550 nm | 0.8 mA/cm$^2$ (−3 V) | 0.79 A/W | 9 GHz | [80] |
| 2020 | InP | MUTC | Ge/GaAs/ InAlAs | InGaAs | - | 1550 nm | 100 mA/cm$^2$ (−3 V) | 0.78 A/W (−4 V) | 28 GHz (−3 V) 40 Gbit/s | [36] |
| 2018 | GaSb | VPIN | GaSb/AlSb | InAs/GaSb | - | 5.5 μm | 10 A/cm$^2$ (−1 V) | - | - | [63] |
| 2019 | GaSb | VPIN | GaSb/AlSb | InAs/GaSb | ~$2.6 \times 10^8$ cm$^{-2}$ | 3–6 μm | 2.3 A/cm$^2$ (−0.1 V, 70 K) | ~1.2 A/W (−0.1 V, 70 K) | - | [64] |
| 2018 | GaSb | nBn | GaSb | InAs/InAsSb | ~$10^9$ cm$^{-2}$ | 2–5 μm | 1 mA/cm$^2$ (−0.5 V, 150 K) | - | - | [81] |
| 2019 | GaSb | nBn | GaSb/AlSb | InAs/InAsSb | $3 \times 10^7$ cm$^{-2}$ | 2.5–6 μm | 1.2 mA/cm$^2$ (−0.1 V, 160 K) | 0.88 A/W (−0.1 V, 200 K) | - | [65] |
| 2019 | GaSb | nBn | Ge/GaAs/ GaSb | InAsSb | - | 4.0–4.2 μm | 50 μA/cm$^2$ (−0.2 V) | - | - | [82] |
| 2020 | GaSb | nBn | GaSb/AlSb | InAs/InAsSb | $3 \times 10^7$ cm$^{-2}$ | 2.5–6 μm | 11.4 μA/cm$^2$ (−0.1 V, 110 K) | 0.88 A/W (−0.1 V, 200 K) | - | [34] |

## 3. Selective Heteroepitaxy of III–V PDs on Si

In contrast to blanket heteroepitaxy, selective heteroepitaxy performs III–V growth in localized areas with large aspect ratios and produces III–V crystals with a variety of geometries. As the generated TDs are often 60° oriented and glide along the {111} plane, they would terminate at the dielectric mask and thus render the epitaxial III–V material TD free. In an analogy to the V-groove technique used in blanket heteroepitaxy, selective heteroepitaxy often initiates nucleation from anisotropic-etched {111} facets and thus completely prohibits the generation of APBs. Unlike the blanket heteroepitaxy method where only GaAs materials could be grown in low TD density, the unique defect trapping mechanism of selective heteroepitaxy yields various TD-free III–V compounds, including GaAs, InP, InGaAs, InAs, and GaSb [22]. Additionally, selective heteroepitaxy confines the majority of the generated defects right at the III–V/Si interface and, as a result, avoids using an excessively thick buffer layer, which facilitates the efficient light coupling between the epitaxial III–V active devices and the passive Si elements. Therefore, these selectively grown III–V PDs could be seamlessly integrated into the Si photonic platform. Moreover, the light propagation direction in Si waveguides could be made orthogonal to the carrier drift direction in the epitaxial III–V PD, which accordingly enables high-performance PDs with the simultaneous achievement of both high responsivity and large bandwidth. However, some complex and delicate III–V device structures, such as superlattices and quaternary compounds, are challenging to grow using selective heteroepitaxy due to the difficulty in controlling the multi-faceted growth front and the resultant composition inhomogeneity of the epitaxial III–V crystal. Nevertheless, this situation is rapidly improving with the

innovations of growth methods as well as the optimizations of process control technology. In this section, as schematically shown in Figure 9, we first introduce the different selective growth methods in terms of the growth direction of the III–V crystal, then review the most representative device demonstrations, and finally highlight the high-performance Si waveguide-coupled III–V PDs seamlessly integrated with Si photonics. The performance parameters of recent III–V PDs integrated on Si by different selective growth methods can be found in Table 3 at the end of this section.

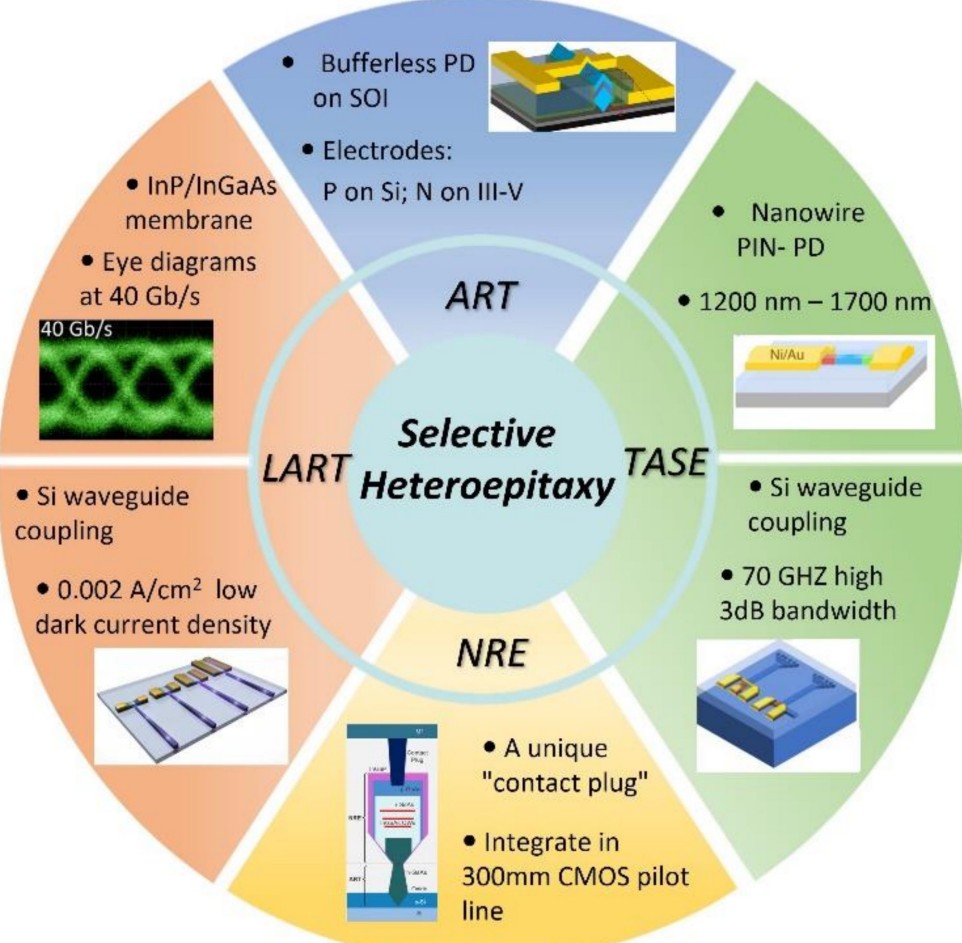

**Figure 9.** Overview of III–V PDs on Si using selective heteroepitaxy. (Blue) Reprinted with permission from Ref. [83]. Copyright 2020, Optical Society of America. (Top green) Reprinted with permission from Ref. [84]. Copyright 2020, The Author(s). (Bottom green)Reprinted with permission from Ref. [27]. Copyright 2022, The Author(s). (Yellow) Reprinted with permission from Ref. [85]. Copyright 2021, IEEE. (Top orange) Reprinted with permission from Ref. [86]. Copyright 2021, Optica Publishing Group. (Bottom orange) Reprinted with permission from Ref. [28]. Copyright 2022, Optica Publishing Group.

### 3.1. Vertically Integrated PDs

#### 3.1.1. Aspect Ratio Trapping

A classical method of selective heteroepitaxy is the "aspect ratio trapping" (ART) technique, which grows III–V materials in high aspect ratio trenches patterned along the [110] direction. Figure 10a,b schematically depicts the defect trapping mechanism of the ART method [21]. When the aspect ratio (AR = h/w) is above 1.4, all TDs originating from the III–V/Si heterogeneous interface would terminate at oxide sidewalls and be trapped, consequently resulting in a TD-free epitaxial layer right above the Si substrate. Planar defects parallel to the trench will also be trapped by the oxide sidewalls, while those

perpendicular to the trench direction would extend across the entire crystal. In addition, the V-groove composed of two etched {111} Si facets avoids the generation of APBs. The ART approach has produced III–V nano-ridges with a variety of compositions and dimensions, and numerous electronic and photonic devices, such as transistors, lasers, modulators, and photodiodes, have been integrated into Si using this method. Figure 10c–e showcases InP and GaAs nano-ridge arrays grown on Si substrates using the ART strategy [87–89].

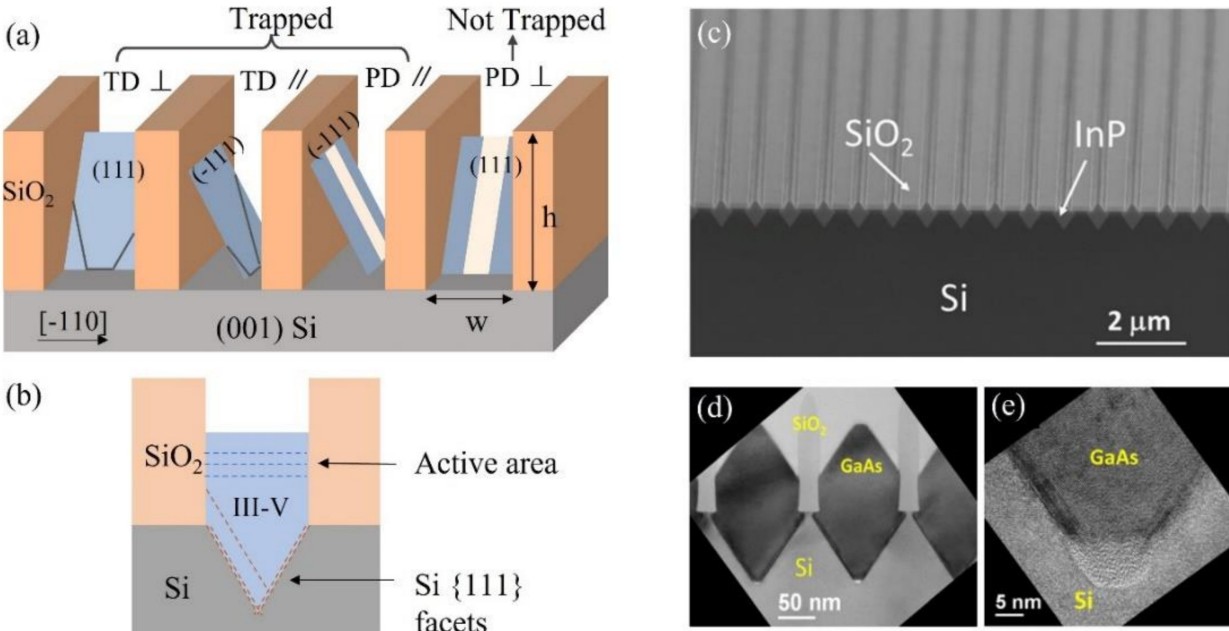

**Figure 10.** (**a**) Defect trapping mechanism of the ART method. (**b**) Schematic diagram of III–V nano-ridges grown in V-shaped grooves. (**c**) SEM image of InP nano-ridges grown on Si substrates using the ART strategy. Reprinted with permission from Ref. [90]. Copyright 2016, AIP Publishing. (**d**) Cross-sectional TEM image of GaAs nano-ridges grown on Si; (**e**) TEM image of the GaAs/Si hetero-interface. Reprinted with permission from Ref. [41]. Copyright 2015, AIP Publishing.

Building from InP/InGaAs nano-ridges by the ART approach, Xue et al. recently demonstrated bufferless III–V PDs directly grown on a V-grooved SOI substrate (see Figure 11a,b) [83]. InP/InGaAs multiple quantum wells were selected as the absorption medium because the nano-ridges feature a convex growth front, and the growth of a thick InGaAs layer could result in composition inhomogeneity. Given the difficulty in patterning the metal pads in the vertical p-i-n III–V nano-ridges, the authors formed the p electrodes on top of the Si device layer while creating the n electrodes atop the n$^+$ InP layer. As shown by the schematic in Figure 11a, such a design ensures that the photogenerated electrons and holes are separated by the built-in electric field and harvested by the two metal electrodes. These PDs exhibit a high responsivity of 1.06 A/W at 1.55 μm (Figure 11c) and operate in the wavelength range from E-band to L-band. However, the large n electrodes atop the nano-ridge induce optical absorption and impair performance. Moreover, the devices can only operate in the MHz range due to the large impedance of the defective III–V/Si interface populated by planar defects. Nevertheless, the "bufferless" feature of the InP/InGaAs PDs promises efficient light coupling between the epitaxial III–V PDs and the Si waveguides, and the authors have theoretically studied the waveguide coupling strategies in another publication [91].

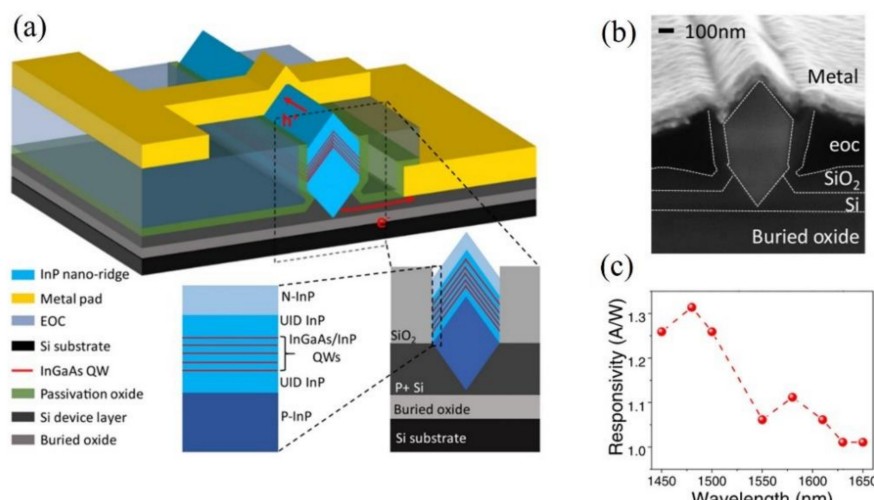

**Figure 11.** (**a**) Schematic of InP/InGaAs PDs grown on (001) SOI using the ART approach. (**b**) 70°
tilted-view SEM image of fabricated PD; dotted lines represent the boundaries of different regions.
(**c**) The evolution of responsivity as a function of wavelength. Reprinted with permission from
Ref. [83]. Copyright 2020, Optical Society of America.

### 3.1.2. Nano-Ridge Engineering

Nano-ridge engineering (NRE) is another effective epitaxy method developed based
on ART. The defect confinement mechanism of NRE is identical to that of ART, but NRE
grows III–V crystals out of the narrow oxide trenches and expands them into larger air-
cladded nano-ridges, as schematically shown in Figure 12a. Due to the large aspect ratio of
the narrow oxide trench, TDs could be completely confined within the oxide trench, and
III–V materials outside the trench become TD free. In addition, the nano-ridges could also
be modified into various shapes by tuning the growth parameters. As evidenced by the
SEM images in Figure 12b,c, researchers have demonstrated box-shaped and triangular-
shaped GaAs nano-ridges by adjusting the V/III ratio, growth rate, and reactor pressure.
GaAs, InGaAs, and GaSb nano-ridges have been grown using the NRE method, and the
measured TD density is about $10^5$ cm$^{-2}$ when the trench opening is less than 100 nm [21,92],
suggesting an excellent crystalline quality of the epitaxial III–V material. Optically pumped
GaAs/InGaAs lasers and efficient optical modulators have recently been integrated into
300 mm Si wafers using the NRE method [93].

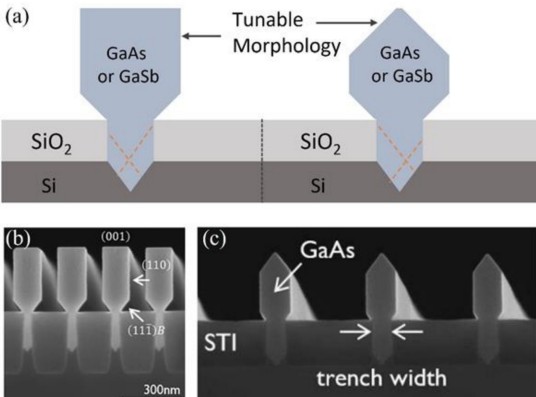

**Figure 12.** (**a**) Schematic of the NRE strategy. (**b**) SEM images of box-shaped GaAs nano-ridges.
Reprinted with permission from Ref. [94]. Copyright 2016, AIP Publishing. (**c**) SEM pictures of
triangle-shaped GaAs nano-ridges. Reprinted with permission from Ref. [95]. Copyright 2016, The
Electrochemical Society.

Building from box-shaped GaAs/InGaAs nano-ridges by the NRE method, Cenk et. Al demonstrated III–V PDs in a 300 mm CMOS pilot line [85]. The active region of this detector consists of a PIN-structured InGaAs/GaAs multiple quantum well, as shown in Figure 13a. It is noteworthy that the nano-ridge PDs feature a unique "contact plug" element to minimize the optical absorption by the metal top contact. Both simulations and experiments demonstrate the improvement of responsivity by the contact plugs. Figure 13b showcases the responsivity of the device under different bias voltages and p-contact plug pitches. Increasing contact plug spacing leads to the expansion of internal responsivity and internal efficiency. The 4.8 μm plug pitch devices exhibit internal responsivity of 0.65 A/W at 1020 nm laser wavelength. The PDs also exhibit a low dark current density of $1.98 \times 10^{-8}$ A/cm$^2$, which speaks to the excellent quality of the epitaxial material and the effectiveness of the InGaP passivation layer. As for future light coupling with Si waveguides, the authors also published an article showing their designs and theoretical calculations of adiabatic tapers [96].

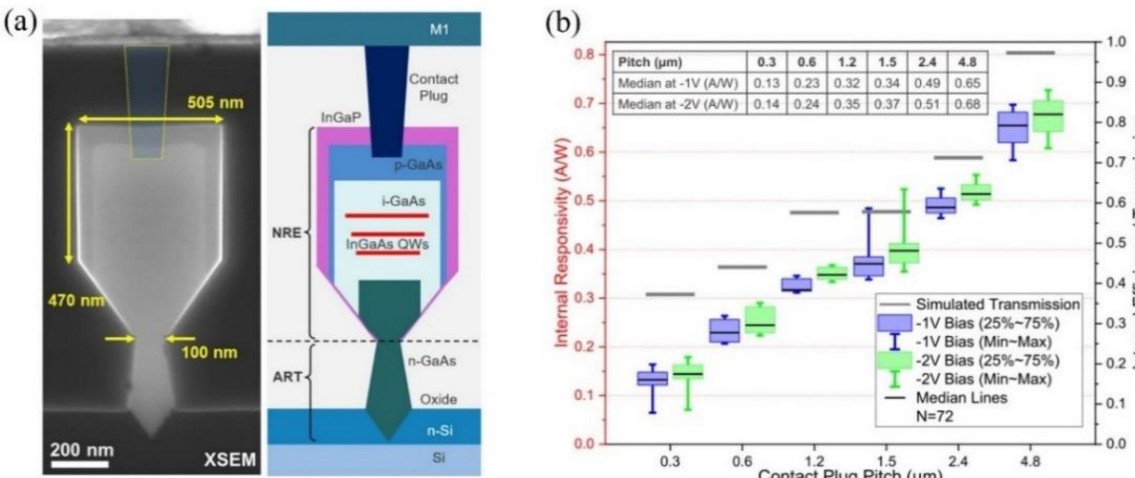

**Figure 13.** (**a**) Cross-section SEM images of GaAs/InGaAs multiple quantum well PD integrated into a 300-mm Si pilot line (left); Schematic of nano-ridge PD cross-section (right). (**b**) Responsivity distributions for different biases and p-contact plug pitches of devices. (grey bars) Simulated transmission values for different p-contact plug pitches, scaled to the right vertical axis. (inset) Table of median responsivity values. Reprinted with permission from Ref. [85]. Copyright 2021, IEEE.

### 3.2. Laterally Integrated PDs

#### 3.2.1. Template-Assisted Selective Epitaxy

In addition to vertical integration schemes such as ART and NRE, researchers have also developed lateral integration strategies. One is called template-assisted selective epitaxy (TASE), which enables the direct growth of in-plane III–V nanostructures from the Si waveguides. As indicated by the schematic in Figure 14a, TASE starts III–V nucleation from a tiny silicon seed, and the evolution of the epitaxial III–V alloy is guided by a hollow oxide template. Since the nucleation region is less than 100 nm in diameter, the release of mismatch stress shifts from plastic relaxation to elastic relaxation, and both TDs and APBs are avoided [97]. In addition, the oxide template can be patterned into various geometries, and the epitaxial III–V compounds accordingly evolve into numerous shapes, such as nanowires, nano-sheets, and micro-disks. Figure 14b,d shows horizontal InGaAs nanowires grown from patterned Si nanowires on the SOI substrate [97]. The TASE approach has also demonstrated the integration of multiple optoelectronic devices on Si. Examples include tunneling field effect transistors, micro-disk lasers, photonic crystal cavity lasers, and high-speed photodiodes [98–102].

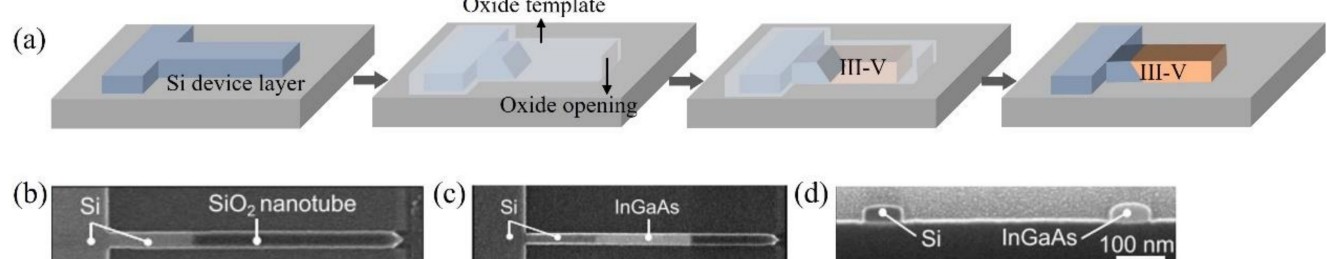

**Figure 14.** (**a**) Schematic of the TASE approach. (**b**–**d**) SEM images of an InGaAs nanowire grown from a Si nanowire seed using the TASE method. Reprinted with permission from Ref. [97]. Copyright 2015, AIP Publishing.

Free-Space Coupled PDs

Using the TASE approach, Svenja et al. have recently demonstrated monolithically integrated InGaAs p-i-n nanowire PDs on Si [84]. Figure 15a schematically presents the growth and fabrication of the III–V nanowire PDs on SOI. With free-space coupling from fibers shown in Figure 15b, the PDs show the response in multiple telecommunication bands from 1200 to 1700 nm with a maximum responsivity of up to 0.68 A/W at 1346 nm. A detection bandwidth of over 25 GHz using 32 Gb/s high-speed optical data reception was also demonstrated (see Figure 15c), the highest reported value for nanostructured PDs at that time. The total footprint of the device is as small as 0.06 $\mu m^2$ due to the nanowire structure of the epitaxial material, which in turn results in an ultra-low capacitance essential for high-speed operation. Besides, the devices emit light at about 1600 nm when forward biased as a light emitting diode.

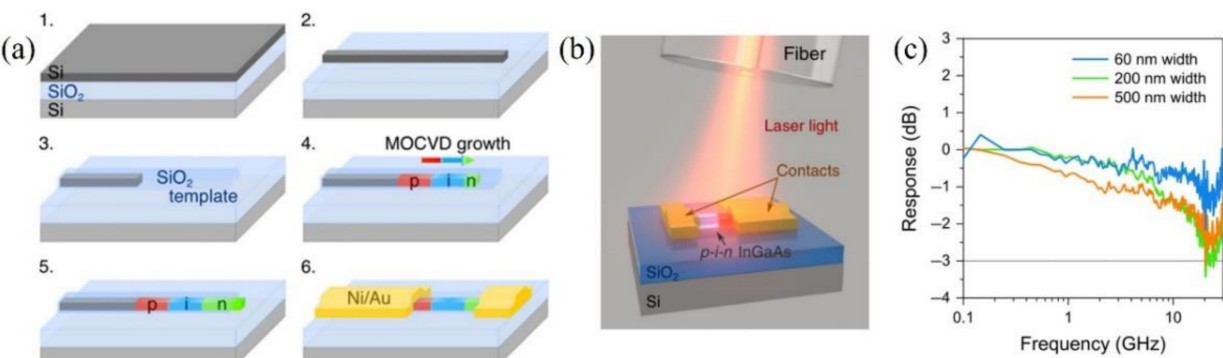

**Figure 15.** (**a**) Schematic of monolithically integrated InGaAs p–i–n nanowire PDs on Si by TASE. (**b**) Schematic of the free-space coupling for PD. (**c**) Response for devices at different widths; the gray line marks the 3dB limit. Reprinted with permission from Ref. [84]. Copyright 2020, The Author(s).

Si Waveguide-Coupled PDs

Following the demonstration of free-space coupled III–V nanowire PDs, researchers from the same group later demonstrated Si waveguide-coupled III–V nanowire PDs by TASE [27]. The device layers are structured sequentially as n-lnP, i-InGaAs, p-lnP, and p-InGaAs as shown by the schematics in Figure 16a. Two different types of waveguide coupling designs, namely the straight device (see Figure 16b) and T-shaped (see Figure 16c) device, are investigated in this study. In straight devices, the Si waveguide also serves as the seed layer for III–V epitaxial, while in T-shape devices, the Si waveguide is separately fabricated and positioned perpendicular to the epitaxial InGaAs layer. In the straight design, metal contacts are patterned atop the epitaxial III–V nanowire and overlap with the light propagation from the Si waveguide, which may induce significant optical absorption loss. In the alternative T-shaped design, light couples vertically from the Si waveguide into the i-region, and the propagating light avoids passing through the metal contacts or

the defective III–V/Si interface. Another advantage of the T-shaped design is that the light propagation is perpendicular to the carrier drift direction, so both high responsivity and large bandwidth can be obtained. Given these reasons, the authors conclude that the T-shaped device is a preferred design. These T-shaped waveguide-coupled PDs feature a dark current density of 0.048 A/cm$^2$ at $-1$ V, and a responsivity up to 0.2 A/W at $-2$ V. High-speed detection with a bandwidth of more than 70 GHz was achieved at 1320 nm, which enables 50 GB of data transmission using on-off keying (OOK) and four-level pulse-amplitude modulation (PAM4). These demonstrations highlight the great potential of the TASE for nanoscale photonic devices in future densely integrated Si photonics.

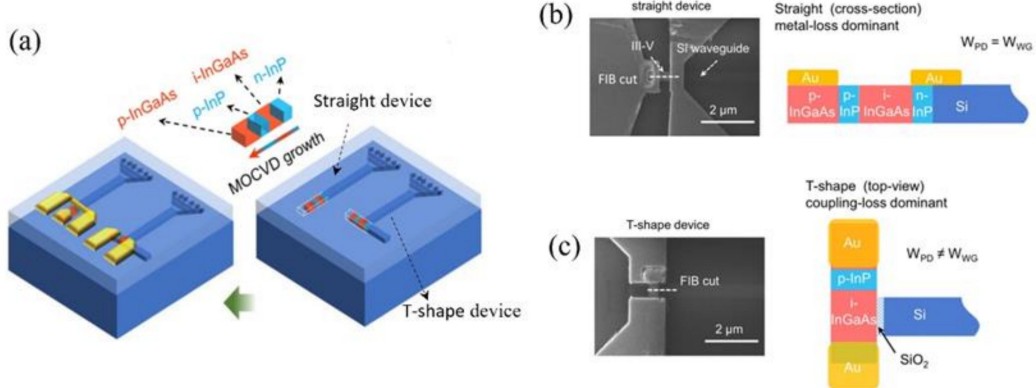

**Figure 16.** (**a**) Schematic of waveguide-coupled III–V PD monolithically integrated on Si. (**b**) Top view SEM image of a straight device (**left**); Schematic of the straight device (**right**). (**c**)Top view SEM of a T-shape device (**left**); Schematic of the T-shape device (**right**). Reprinted with permission from Ref. [27]. Copyright 2022, The Author(s).

### 3.2.2. Lateral Aspect Ratio Trapping

Lateral aspect ratio trapping (LART) combines the advantages of the ART and the TASE methods and manifests unique advantages not available to either of these two technologies. As shown by the schematic in Figure 17a, LART initiates III–V growth inside lateral oxide trenches with a depth of up to 10 μm, and the III–V alloy evolves laterally into a membrane. The ultra-large aspect ratio effectively confines the propagation of TDs, and the anisotropic wet-etched {111} Si facets avoid the formation of APBs in the epitaxial material. In the conventional vertical ART method, the aspect ratio is determined by the depth over the width of the vertical trench, and enlarging the width would severely compromise the defect necking effect. However, in the LART approach, the aspect ratio depends on the width of the epitaxial III–V material over the thickness of the Si device layer; extending the lateral dimension by no means jeopardizes the defect necking effect. As a result, unlike the TASE approach where III–V nanostructures are obtained, the LART method often yields large-dimension III–V membranes, as evidenced by the microscopic and SEM images in Figure 17b–e. These III–V-on-insulator membranes could serve as platforms for the epitaxy/fabrication of a variety of III–V optoelectronic devices. Recently both optically pumped lasers and high-performance PDs have been demonstrated using the LART approach [86,103].

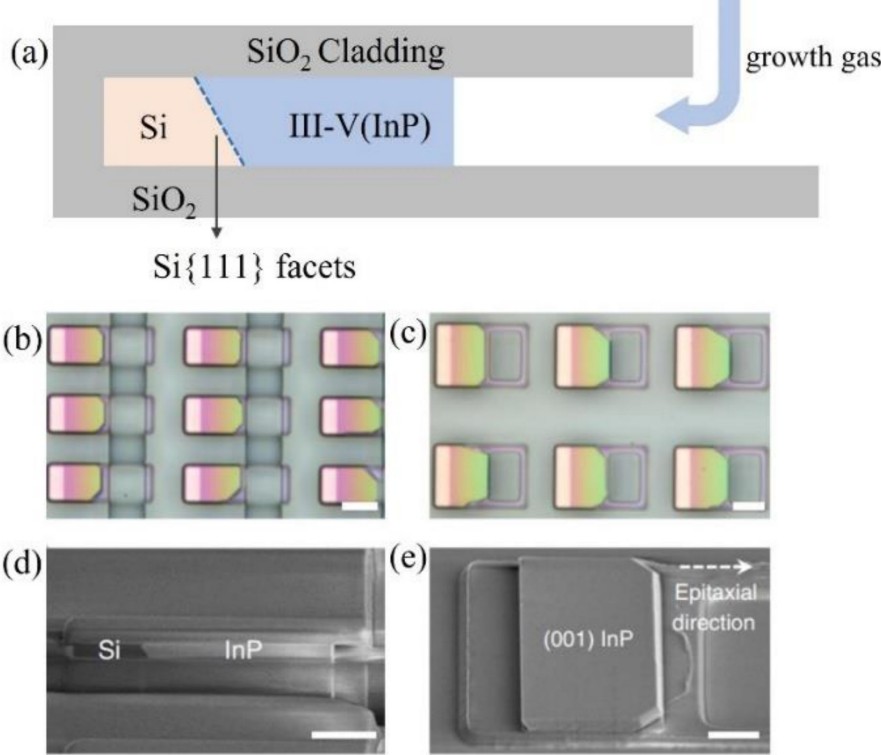

**Figure 17.** (**a**) Schematic of the LART strategy. (**b,c**) Optical microscope images of InP membranes with pattern lengths of 5 µm and 10 µm, respectively. Scale bar, 5 µm. (**d**) Cross-sectional SEM image of the epitaxial InP. Scale bar, 2 µm. (**e**) SEM image of the InP membrane with a pattern length of 10 µm after top oxide and Si removal, Scale bar, 2 µm. Reprinted with permission from Ref. [103]. Copyright 2021, The Author(s).

Free-Space Coupled PDs

Based on InP membranes grown on SOI by the LART approach, Xue et al. present high-performance III–V PDs with different absorption materials and multiple device dimensions (Figure 18a) [86]. Bulk InGaAs and five periods of InGaAs/InP QWs (Figure 18a) are selected as absorption layers in the p-i-n structure, respectively. The width of the grown n-InP and p-InGaAs exceeds 2.0 µm to facilitate metal electrode contact. In addition, device dimensions range from 0.5 µm to 20 µm in length. In order to improve device performance and increase stability, benzocyclobutene (BCB) is used as a planarization layer as well as an effective passivation layer before metal contacts are defined (Figure 18b). The fabricated devices exhibit a 3dB bandwidth over 40 GHz, a 0.3 A/W response at 1550 nm, and a 0.8 A/W response at 1310 nm (Figure 18c). The operating wavelength ranges from 1240 nm to 1650 nm. The dark current of PDs with bulk InGaAs as absorption material is several times larger than those with InP/InGaAs QWs, which probably stems from the structural defects in thick bulk InGaAs layers with composition inhomogeneities. Figure 18d shows eye diagrams for the PDs with a length of 2 µm. The clear open eye diagram at 15 and 40 Gb/s proves the high-speed characteristics of PDs selectively gown on SOI by the LART approach.

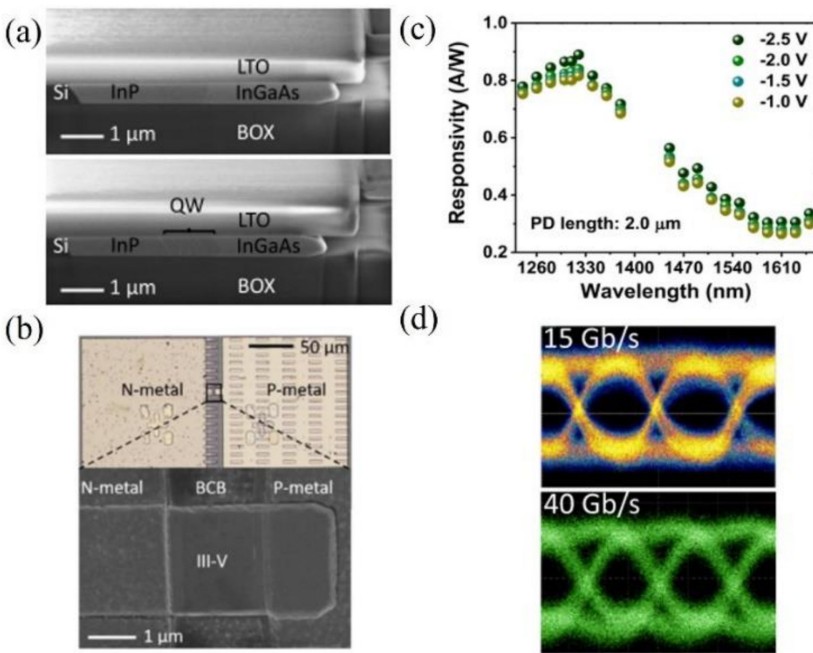

**Figure 18.** (**a**) SEM image of as-grown InP/InGaAs p-i-n PD by LART. (**b**) Top-view optical image of the fabricated III–V PD. (**c**) Responsivity from 1240 nm to 1650 nm. (**d**) Measured eye diagrams at 15 and 40 Gb/s. Reprinted with permission from Ref. [86]. Copyright 2021, Optica Publishing Group.

Si Waveguide-Coupled PDs

In the following years, the team demonstrated Si waveguide-coupled III–V PDs with various dimensions on a standard (001) SOI platform. Figure 19a displays the 3D schematic of the fabricated devices, and Figure 19b,c details the placement of the epitaxial III–V and the Si waveguide in directions parallel and perpendicular to the growth orientation. Given the co-planner configuration of the epitaxial III–V and the Si device layer, butt coupling was selected to decouple light illumination and carrier collection. To accurately align the Si waveguide and the absorption region, the Si waveguide was defined after the heteroepitaxy process. As shown in Figure 20a,b, a unique taper structure was also designed to improve the light coupling efficiency up to more than 70% with a gap between 200 nm and 1.0 μm, which significantly improves the tolerance in the fabrication process. The fabricated Si waveguide-coupled PDs exhibit a low dark current density of 0.002 A/cm$^2$ (Figure 20d), a responsivity of 0.4 A/W at 1 V, 0.4 A/W at 1.3 μm, 0.2 A/W at 1.5 μm, a saturation photocurrent of over 1 mA, and an operating wavelength over the entire telecom band. In addition, a 3dB bandwidth beyond 52 GHz and data communications over 112 Gb/s were also achieved, as shown in Figure 20c. The high-performance Si waveguide-coupled III–V PDs highlight the potential of the LART method in integrating III–V photonic devices in the Si photonics platform.

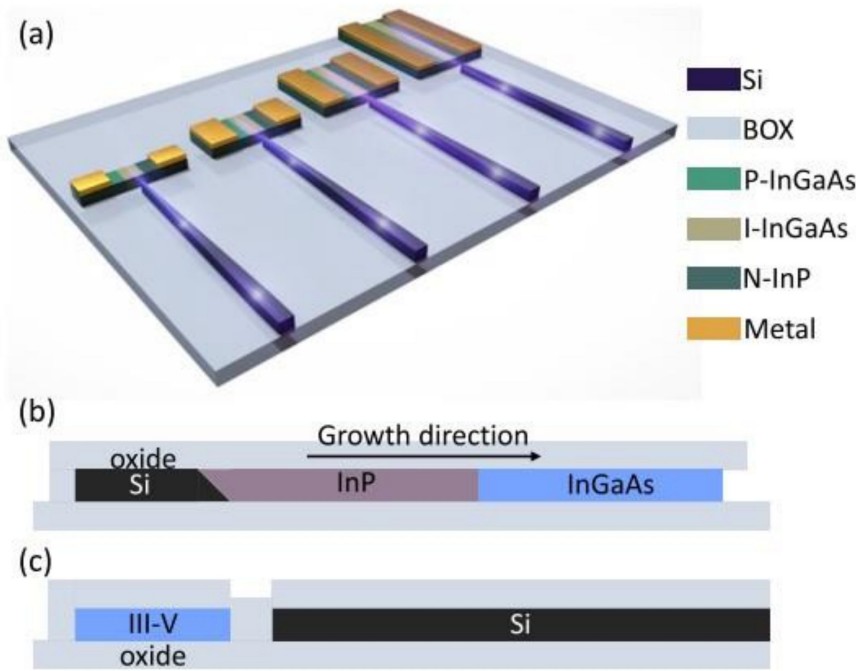

**Figure 19.** (**a**) 3D schematic diagram of Si waveguide-coupled III–V PDs with different sizes integrated on SOI. (**b**) Schematic of lateral epitaxy of PD device layer. (**c**) Cross-section schematic of placement of the epitaxial III–V and the Si waveguide. Reprinted with permission from Ref. [28]. Copyright 2022, Optica Publishing Group.

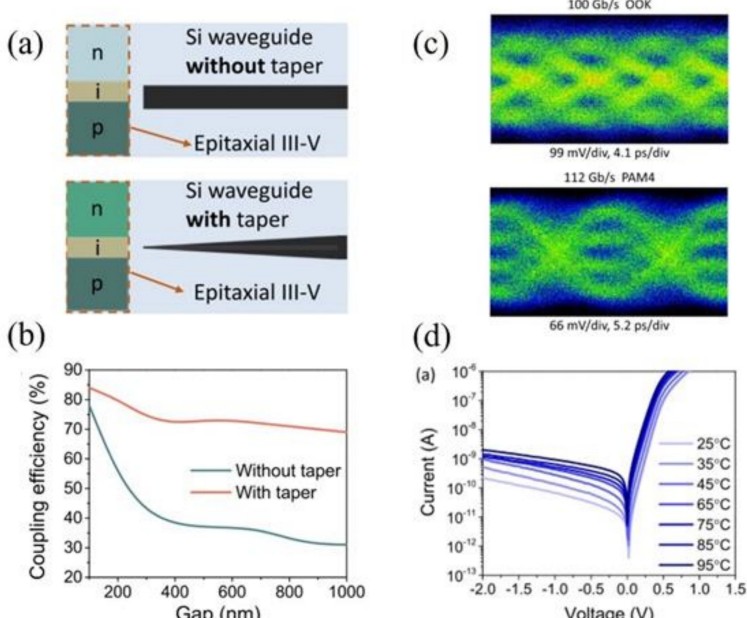

**Figure 20.** (**a**) Schematic of the coupling strategy utilizing direct butt-coupling and Si inverse taper. (**b**) Simulated coupling efficiencies between Si waveguides and III–V PDs at various gaps. (**c**) 100 Gb/s OOK eye diagram measured at −2 V (**top**); 112 Gb/s PAM4 eye diagram under −2 V bias (**bottom**). (**d**) Dark current from 25 °C to 95 °C. Reprinted with permission from Ref. [28]. Copyright 2022, Optica Publishing Group.

**Table 3.** Summary of selective heteroepitaxy III–V PDs.

| Year | Device Integration | Substrate | Material | Wavelength (nm) | Dark Current | Dark Current Density | Responsivity | Speed (3 dB Bandwidth) | Waveguide Coupled | Refs. |
|------|-------------------|-----------|----------|-----------------|--------------|----------------------|--------------|------------------------|-------------------|-------|
| 2020 | ART | SOI | InP/InGaAs | 1450–1650 | 16 nA (−0.5 V) | 33 mA/cm$^2$ | 1.06 AW$^{-1}$ (1.55 μm) | - | No | [83] |
| 2021 | ART + NRE | (001) Si | GaAs /InGaAs | - | <1 pA | $1.98 \times 10^{-8}$ A/cm$^2$ | 0.65 AW$^{-1}$ (1020 nm) | 1.1–1.9 GHz | No | [85] |
| 2020 | TASE | SOI | InGaAs | 1200–1700 | 1.7 nA (−2 V) | - | 0.68 AW$^{-1}$ (1346 nm) | 25 GHz | No | [84] |
| 2022 | TASE | SOI | InP/InGaAs | 1200–1700 | 0.04 nA (−1 V) | 0.048 A/cm$^2$ | 0.6 AW$^{-1}$ (1310 nm) 0.2 AW$^{-1}$ (1550 nm) | 70 GHz | Si | [27] |
| 2021 | LART | SOI | InP/InGaAs | 1240–1650 | 0.55 nA | - | 0.3 AW$^{-1}$ (1550 nm) 0.8 AW$^{-1}$ (1310 nm) | 40 GHz | No | [86] |
| 2022 | LART | SOI | InP/InGaAs | 1260–1650 | 60 pA | 0.002 A/cm$^2$ | 0.2 AW$^{-1}$ (1.5 μm) 0.4 AW$^{-1}$ (1.3 μm) | 52 GHz | Si | [28] |

## 4. Summary and Perspective

### 4.1. Summary

In this article, we reviewed recent progress in III–V PDs monolithically integrated on Si using two approaches: blanket heteroepitaxy and selective heteroepitaxy. Blanket heteroepitaxy of GaAs-based PDs has extended their detection wavelength to the O band and the mid-infrared band thanks to the incorporation of self-assembled InAs QDs. An ultra-low dark current density is also achieved owing to the low TD density of $10^6$ cm$^{-2}$. Taking advantage of the unique MUTC structures, the telecom InP-based PDs feature both high-speed and high-power handling capabilities. For GaSb-based PDs in the mid-infrared band, the dark current has also been drastically reduced thanks to the unique nBn structure. Despite these advances, the TD density of InP and GaSb thin films grown on Si needs to be progressively reduced to further suppress the dark current density. In addition to the experimental designs, efficient light coupling strategies between the epitaxial III–V PDs and Si waveguides must be experimentally verified.

Selective heteroepitaxy has enabled the monolithic integration of high-performance Si waveguide-coupled III–V PDs on Si photonics platforms. The unique defect necking effect not only produces TD-free III–V materials that result in ultra-low dark current density but also confines the defective region at the III–V/Si interface, which leads to efficient light coupling from Si waveguides. Benefiting from innovations in epitaxial techniques like NRE, TASE, and LART, high-performance III–V PDs with dark current density down to $1.98 \times 10^{-8}$ A/cm$^2$, 3dB bandwidth over 70 G, and data transmission over 100 G have been experimentally demonstrated.

### 4.2. Perspective

In an analogy to Ge PDs selectively grown on SOI using chemical vapor deposition in present Si photonics foundries, the future integration of III–V photonic devices into Si photonics platforms would probably adopt a similar manner, namely selective heteroepitaxy using metal organic chemical vapor deposition (MOCVD). This monolithic strategy is widely considered to feature the lowest cost, highest throughput, and largest integration density. However, to meet the stringent requirements in practical applications, the selectively grown III–V PDs must feature low dark current, high speed, large responsivity, and wide operation wavelength range. In addition, the entire process must be compatible with the existing procedures in Si photonics foundries. Careful handling of the thermal budget and contamination is mandatory.

Since the selectively grown III–V crystals often feature large surface-to-volume ratios and multiple facets/interfaces, effective passivation techniques using dielectric or regrown III–V layer must be developed to reduce the surface leakage current and thus minimize the dark current density. Current III–V UTC PDs grown on native InP substrate could operate well beyond 300 G [104], which suggests a huge room for improvement for III–V PDs selectively grown on Si. Achieving homogeneous quaternary compounds on multi-faceted III–V crystals and designing efficient UTC structures could further boost the high-speed performance of III–V PDs on Si using selective heteroepitaxy. Although efficient butt coupling strategies have been experimentally verified, the coupling efficiency and resultant responsivity could be further improved using evanescent coupling. In addition, evanescent coupling could minimize unwanted reflections from the III–V/Si interface as well as increase the power handling capabilities of the epitaxial III–V PDs.

Until now, III–V PDs by selective heteroepitaxy were all fabricated from simple p-i-n structures with bulk alloy or quantum wells as the absorption medium. In addition, only GaAs and InP-based PDs have been experimentally demonstrated. As a result, the detection wavelength is restricted to the near-infrared and the telecom band. Adopting InAs QDs as the absorption medium or selectively growing GaSb/InAs-related compounds on Si could extend the operating wavelength to the mid-infrared region. Apart from conventional p-i-n junctions, growing complex PD structures such as APDs, UTC PDs, and nBn PDs could unleash the full potential of selective heteroepitaxy. Together with the integration of III–V lasers, the monolithic integration of III–V PDs on Si photonics platforms using direct heteroepitaxy will soon move from laboratories in academia to production lines in the industry.

**Author Contributions:** Writing—original draft preparation, C.Z. and D.F.; writing—review and editing, C.Z., D.F., Y.J. and Y.H.; supervision, Y.H. All authors have read and agreed to the published version of the manuscript.

**Funding:** This research was funded by National Natural Science Foundation of China (Grant No. 41030112), Guangdong Natural Science Foundation (Grant No. 42030052) and Guangzhou Municipal Science and Technology Project (Grant No. 42050079).

**Institutional Review Board Statement:** Not applicable.

**Informed Consent Statement:** Not applicable.

**Data Availability Statement:** Not applicable.

**Conflicts of Interest:** The authors declare no conflict of interest.

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
