# Peer review of "Recent Progress in III–V Photodetectors Grown on Silicon"

_photonics, doi:10.3390/photonics10050573_

Round 1

Reviewer 1 Report

This review paper is well-organized and well-written, an extensive review of the most recent advances in III-V PDs directly grown on Si using two different epitaxial techniques has been done. 

Reviewer 2 Report

In this article, authors have reviewed the recent progress in III–V PDs monolithically integrated on Si using two approaches: namely, blanket heteroepitaxy and selective heteroepitaxy. Past relevant reports are critically reviewed and appropriately presented. However, there are still couple of points authors should elaborate before it is accepted as publication in the PHOTONICS.

1.      Fundamental background of the study is not well treated in introduction.

2.      More recent studies from 2022 and 2023 are required to add.

3.  Add a section discuss little bit about the working principles of III-V PDs and incorporation with Si.

4.      Most of text in Figure 1, 2 and 5 is not clear.

5.      paragraphs are too long, make reasonable small paragraphs.

6.      Grammatical errors and typos should be corrected.

Reviewer 3 Report

In this article , the authors review the most recent advances in III-V PDs directly grown on Si using two different epitaxial techniques: blanket heteroepitaxy and selective heteroepitaxy.

The article is interesting and well written . However, some issues should be addressed.

1)      In the introduction, the authors should clearly highlight the novelty and the importance of their work in comparison to previous works. 

         2) The abstract should be improved. The authors should highlight  in the abstract the novelty of their review .  

3) In the introduction the authors should refer to recent advances about silicon photonics.

See for example:

·       Free-Space Applications of Silicon Photonics: A Review. Micromachines 202213, 990. https://doi.org/10.3390/mi13070990

·       Modulators in Silicon Photonics—Heterogenous Integration & and Beyond. Photonics 20229, 40. https://doi.org/10.3390/photonics9010040

·        An Optimization Framework for Silicon Photonic Evanescent-Field Biosensors Using Sub-Wavelength Gratings. Biosensors 202212, 840. https://doi.org/10.3390/bios12100840

·       Cyclic Voltammetry and Impedance Measurements of Graphene Oxide Thin Films Dip-Coated on n-Type and p-Type Silicon. Crystals. 2023; 13(1):73. https://doi.org/10.3390/cryst13010073

4) The authors should improve the future prospective section of their review.

Round 2

Reviewer 3 Report

I recommend the publication of this article